# Transcriptome Analysis Reveals Key Genes Involved in the Response of *Pyrus betuleafolia* to Drought and High-Temperature Stress

**DOI:** 10.3390/plants13020309

**Published:** 2024-01-20

**Authors:** Panpan Ma, Guoling Guo, Xiaoqian Xu, Tingyue Luo, Yu Sun, Xiaomei Tang, Wei Heng, Bing Jia, Lun Liu

**Affiliations:** College of Horticulture, Anhui Agricultural University, Hefei 230036, China; 13966280342@163.com (P.M.); glintguo@163.com (G.G.); 13775485346@163.com (X.X.); luotingyue2023@163.com (T.L.); ahausunyu@163.com (Y.S.); tangxiaomei@ahau.edu.cn (X.T.); hengwei@ahau.edu.cn (W.H.)

**Keywords:** *Pyrus betuleafolia*, transcriptome analysis, drought, high temperature, resistance

## Abstract

Drought and high-temperature stress are the main abiotic stresses that alone or simultaneously affect the yield and quality of pears worldwide. However, studies on the mechanisms of drought or high-temperature resistance in pears remain elusive. Therefore, the molecular responses of *Pyrus betuleafolia*, the widely used rootstock in pear production, to drought and high temperatures require further study. Here, drought- or high-temperature-resistant seedlings were selected from many *Pyrus betuleafolia* seedlings. The leaf samples collected before and after drought or high-temperature treatment were used to perform RNA sequencing analysis. For drought treatment, a total of 11,731 differentially expressed genes (DEGs) were identified, including 4444 drought-induced genes and 7287 drought-inhibited genes. Kyoto Encyclopedia of Genes and Genomes (KEGG) analysis revealed that these DEGs were more significantly enriched in plant hormone signal transduction, flavonoid biosynthesis, and glutathione metabolism. For high-temperature treatment, 9639 DEGs were identified, including 5493 significantly upregulated genes and 4146 significantly downregulated genes due to high-temperature stress. KEGG analysis showed that brassinosteroid biosynthesis, arginine metabolism, and proline metabolism were the most enriched pathways for high-temperature response. Meanwhile, the common genes that respond to both drought and high-temperature stress were subsequently identified, with a focus on responsive transcription factors, such as MYB, HSF, bZIP, and WRKY. These results reveal potential genes that function in drought or high-temperature resistance. This study provides a theoretical basis and gene resources for the genetic improvement and molecular breeding of pears.

## 1. Introduction

Drought and high temperature, the most common environmental stressors, are predominant abiotic stresses in global agriculture that seriously affect crop growth and development [1,2]. Drought or high-temperature stress causes accumulation of reactive oxygen (ROS), structural damage in the cell membrane, and reductions in intracellular enzyme activity in the plants [3,4].

In the process of continuously adapting to environmental changes, plants have evolved various ways to cope with drought or heat stress, such as morphology, cell physiology, metabolite changes, and molecular regulation [5]. In mild drought or heat stress, plant stomata close, leaves curl, and they slow down growth to reduce water loss. Severe stress will lead to excessive accumulation of reactive oxygen species (ROS) in the plant, and excessive ROS seriously affects the physiological processes of various cells [6]. In the event of excessive ROS accumulation, plants will activate their defense system, which includes the enzymatic system composed of superoxide dismutase (SOD), catalase (CAT), peroxidase (POD), and the nonenzymatic system, composed of antioxidant substances such as glutathione (GSH) and ascorbic acid (ASA) [7].

Plants also respond to stress at the molecular level through the regulation of stress-related gene expression [1]. Three main types of stress response genes have been reported in plants. The first type of response genes are proteins involved in protein translation or modification, signaling cascades, and transcriptional regulation, such as protein kinase, protein phosphatase, and various transcription factors. The second type of genes is mainly functional proteins that protect cell membranes and other proteins, such as antioxidants and reverse-osmosis proteins. The third type of genes are mainly proteins involved in the uptake and transport of water molecules, ions, and nutrients, such as aquaporins and sugar transporters [1].

Additionally, the function of plant hormone involvement in resistance to abiotic stress has been widely reported. When plants are exposed to drought and high-temperature stress, abscisic acid reduces stomatal opening and deals with stress by regulating the expression of a series of genes [8]. Brassinolide, methyl jasmonate acid, and ethylene have also been reported to be involved in the plant stress response, including high-temperature and drought stress [9,10,11,12]. However, it is not yet clear which of the hormone signaling genes from *Pyrus betuleafolia* are involved in response to drought or high-temperature stress.

RNA sequencing (RNA-seq) has been widely used in the large-scale screening of differentially expressed genes. This technology can quickly reveal the expression patterns of a large number of genes and the information from the corresponding signal pathways [13]. Several studies have used RNA-seq technology to reveal the expression characteristics of a large number of genes in response to drought or high-temperature stress [14]. In *Casuarina equisetifolia*, the study identified a total of 5033 and 8159 differentially expressed genes (DEGs) that respond to drought after 2 h and 24 h, respectively. These DEGs were involved in the transduction of plant hormone signals, jasmonic acid (JA) biosynthesis, flavonoid biosynthesis, and phenylpropanoid biosynthesis [15]. In *Hibiscus cannabinus*, analyzing the transcriptome of kenaf leaves under drought stress, eight transcription factors were detected in the AP2/ERF, MYB, NAC, and WRKY families that were associated with drought stress responses [16].

For high temperatures, in Chrysanthemum, a total of 18,286 DEGs were identified under normal and high-temperature conditions; functional annotation and enrichment analyses revealed that these DEGs were significantly enriched in the heat shock response and flavonoid pathways [17]. In soybeans, the RNA-seq data showed that many drought-response genes were mainly involved in the defense response, the response to biological stimuli, the auxin-activated signaling pathway, the transduction of starch and sucrose metabolism, plant hormone signals, and MAPK signaling pathway [18]. 

Pear is one of the most important temperate fruits in the world. Continuous drought or high-temperature stress seriously affects the growth of pear trees, resulting in early defoliation. These stresses also influence fruit growth, seriously affecting their yield and quality. *Pyrus betuleafolia* is widely used as rootstock in pear production. A previous study has identified dehydration response genes in *Pyrus betulaefolia* by RNA-seq. As a result, a total of 19,532 genes responded to dehydration. Most of these genes were involved in the response to dehydration, metabolism, and signaling of hormones [19]. However, the pear, as a woody plant, is susceptible to drought and high-temperature stress. The mechanism of the *Pyrus betuleafolia* response to drought or high-temperature stress is still unclear. 

In our study, *Pyrus betuleafolia* seedlings with resistance to drought and high temperatures were screened, respectively. Then, drought- or high-temperature-resistant seedlings were treated with drought or high temperature, respectively, and their responses to the stresses were monitored at different time points. The samples at 0 h and 48 h after drought treatment and the samples at 0 h and 9 h after high-temperature treatment were used to perform the RNA-seq, respectively. As a result, a large number of genes that respond to drought and/or high temperatures were selected. In this study, we analyzed common genes that respond to both drought and high temperature and enriched the pathways associated with DEG. This study will provide the basis for the analysis of the molecular mechanism of pear drought and high-temperature resistance along with the theoretical basis and gene resources for the genetic improvement and molecular breeding of pears. 

## 2. Results

### 2.1. Detection of Antioxidant Enzyme Activity in Resistant Pyrus betuleafolia Seedlings after Drought or High-Temperature Treatment

We selected 20-day-old drought-resistant (DR) plants, with 2–3 green and stretched leaves, in 324 *Pyrus betulaefolia* seedlings that come from 8 major pear cultivation regions (Figure 1A,B). These DR seedlings were subjected to PEG (20%) drought treatment. We collected the samples 0 h, 12 h, 24 h, 36 h, 48 h, 60 h, 72 h, and 84 h after drought treatment, respectively. Then, we detected the SOD activity, peroxidase (POD) activity, and catalase (CAT) activity of all of the samples. The results showed that the SOD, POD, and CAT activities of samples increased significantly at 48 h, 60 h, and 72 h compared to the plants before treatment. Since the increase rate of enzyme activity was the largest in the 48 h sample. A duration of 48 h after drought is likely the point at which most of the seedlings’ genes respond to drought stress. Therefore, we selected the 0 h and 48 h samples after drought treatment to perform the RNA-seq. However, there is the possibility that additional genes respond to drought but do so at earlier or later time points than what were tested in this study.

Meanwhile, we screened high-temperature-resistant (HTR) seedlings, with 2–3 green and stretched leaves, among 289 *Pyrus betulaefolia* seedlings from 8 major pear cultivation regions. Then, these HTR seedlings were subjected to temperatures of 47 °C in the growth chamber. We collected the samples at 0 h, 3 h, 6 h, 9 h, 12 h, and 18 h after treatment at 47 °C. Further, we detected SOD, POD, and CAT activity of all the samples. The results showed that the SOD, POD, and CAT activity of samples increased significantly at 6 h, 9 h, 12 h, and 18 h compared with the plants before treatment. Furthermore, the increase in enzyme activity was the largest in the 9 h sample, implying that 9 h after high-temperature treatment is probably the time point at which most of the genes in the seedlings respond to drought stress. Thus, we selected the 0 h and 9 h samples to perform the RNA-seq. However, there is the possibility that additional genes respond to high temperature but do so at earlier or later time points than those that were tested in this study.

### 2.2. Transcriptome Analysis of Pyrus betulifolia in Response to Drought or High-Temperature Stress

For drought treatment, a total of 268,734,008 clean reads were obtained through the RNA-seq, and GC content ranged from 47.56% to 47.83%. High-quality reads of all the libraries that were obtained, with the Q30 base percentage of each library being >91%, indicated that each library could be used for further analysis. In this study, principal component analysis (PCA) was used to reduce the dimension and to analyze the correlation of all the samples. For the drought, RNA-seq, the first principal component (PC1) explained 71.06%, while the second principal component explained 18.53% (Figure 2A). A total of 11,731 DEGs were identified in the drought treatment group compared to the control group, including 4444 significantly upregulated genes and 7287 significantly downregulated genes (Figure 2B). The expression patterns of the DEGs with three independent biological replicates for each time point were visualized with a heat map (Figure 2C). The results showed that DEGs are up- or downregulated by drought. Additionally, the *k*-means clustering analysis was performed to understand the general trend of these DEGs. The top two clusters with the most DEGs are shown in Figure 2D; the expression of those DEGs in cluster 1 was higher in the drought treatment group at 48 h than at 0 h. These DEGs in cluster 2 were significantly negatively regulated by drought treatment (Figure 2D).

For high-temperature treatment, the RNA-seq data generated 256,452,616 clean reads, and the GC content ranged from 47.56% to 47.83%. High-quality reads of all libraries were obtained, with the base percentage of each library being >93%. PC1 explained 65.11% and the PC2 explained 16.48% (Figure 2E). Compared to the control group, the RNA-seq data identified 9639 DEGs in the high-temperature treatment group, including 5493 significantly upregulated genes and 4146 significantly downregulated ones (Figure 2F). All DEGs with three independent biological replicates for each time point were visualized with a heat map (Figure 2G). These results showed that DEGs upregulate or downregulate with high temperatures (Figure 2G). Additionally, the *k*-means cluster analysis showed that the top three clusters with the most DEGs were shown in Figure 2H. The expression of those DEGs in cluster 1 was lower in the high-temperature treatment group at 9 h than at 0 h. These DEGs in clusters 2 and 3 were significantly upregulated by high-temperature treatment (Figure 2H).

In order to reveal the pathways of differentially accumulated metabolites and classify the functions of the DEGs, we performed an enrichment analysis of the Gene Ontology (GO) and Kyoto Encyclopedia of Genes and Genomes (KEGG) pathways. For the GO categories, the DEGs of drought treatment were annotated with 20 biological processes, and most genes located in the “photosynthesis”, “phosphotransferase activity”, “alcohol group”, “protein phosphorylation”, “protein kinase activity”, “signal transduction”, and “defense responses” (Figure 3A). KEGG enrichment analyses found that the “plant hormone signal transduction”, “flavonoid biosynthesis”, “glutathione metabolism”, and “glycolysis/gluconeogenesis” were overrepresented by the annotation analysis (Figure 3B). 

For the high-temperature treatment, the GO analysis revealed that these DEGs were mostly categorized into “structural molecule activity”, “cellular amide metabolic process”, “structural ribosome constituent”, “peptide biosynthetic process”, “translation”, and “ribosomal subunit” (Figure 3C). KEGG enrichment analyses revealed that the main four pathways were “brassinosteroid biosynthesis”, “arginine and proline metabolism”, “purine metabolism”, and “glycerolipid metabolism”, which were significantly related to the high-temperature response (Figure 3D). In addition, heat-shock proteins (HSP) were reported to be involved in resistance to the high-temperature stress [20,21,22]. In this study, we identified 56 HSPs that significantly responded to high-temperature stress in RNA-seq data (Appendix A).

### 2.3. Plant Hormone Signal Transduction Pathways Were Involved in Response to Drought Stress

Based on KEGG enrichment analysis, the plant hormone signal transduction pathway was closely related to drought stress. We investigated these DEGs that responded to drought and were enriched for the transduction of plant hormone signals in the RNA-seq data of drought treatment. As a result, the hormone signal transduction pathways, including cytokines, gibberellin, abscisic acid, brassinosteroid, ethylene, salicylic acid, JAs, and auxin, were involved in drought response in DR. For the cytokine signal transduction pathway, one DEG that encoded *Cytokinin Response 1 (CRE1) proteins* (LOC103935117) and two DEGs that encoded *type-B Arabidopsis Response Regulators* (*B-ARR)* (LOC103953811, LOC103939405) were significantly upregulated (Figure 4); meanwhile, one DEG that encoded *CRE1 proteins* (LOC103943104), four DEGs that encoded *Arabidopsis Histidine Phosphotransfer Protein (AHP)*, and two DEGs that encoded *B-ARR* (LOC103937948, LOC103949872) were significantly downregulated (Figure 4). For the gibberellin signal transduction pathway, DEGs encoding *Gibberellin Insensitive Dwarf1 (GID1) Protein* (LOC103938790) were significantly upregulated by drought, while DELLA proteins and related transcription factors (TF) were significantly downregulated. For the abscisic acid signal transduction pathway, five DEGs encoding *Type 2C protein phosphatases* (*PP2C*) and two DEGs encoding *Sucrose non-fermenting 1 related protein kinases* (*SnRK2*) (LOC103956414, LOC103937906) were significantly increased by drought stress; meanwhile, five DEGs encoding *Pyrabactin resistance 1-like* (*PYR/PYL)* and two DEGs encoding *SnRK2* (LOC103931793, LOC103955653) were significantly decreased (Figure 4).

In the brassinosteroid signal transduction pathway, one DEG encoding *BAK1* (LOC103952612) and three DEGs that encode *brassinosteroid-signaling kinase (BSK) proteins* (LOC103963256, LOC103934486, LOC103932504) were significantly upregulated; meanwhile, two DEGs that encode *BRI1-associated Receptor Kinase 1 (BAK1) Protein* (LOC103939461) and one DEG encoding *Brassinazole-resistant 1 (BZR1/2)* were significantly downregulated. In the ethylene signal transduction pathway, nine DEGs were significantly upregulated, including two DEGs encoding *Ethylene Receptor (ETR)*, one DEG encoding *stress-induced MAPKK (Mitogen-activated Protein Kinases Kinases) (SIMKK)*, five DEGs encoding *EBF1/2*, and one DEG encoding *Ethylene Response Factor (ERF)*, were significantly upregulated (Figure 4). In the auxin signal transduction pathway, the DEGs encoding *Auxin Transporter Protein 1 (AUX1)* (LOC103936499), *Transport Inhibitor Response Protein 1 (TIR1)* (LOC103963931), and *Auxin Response Factor (ARF)* (LOC103929308, LOC103944151, LOC103949259) were significantly downregulated, while the DEGs encoding *ARF* (LOC103941558) and *Gretchen Hagen3* of auxin early response gene (GH3) (LOC103942180, LOC103938212, LOC103950385, LOC103965914, LOC103961617) were significantly upregulated (Figure 4). Many DEGs were also identified as drought-response genes in the JAs and salicylic acid signal transduction pathways in this study. All these results revealed potential key genes that affect *Pyrus betulaefolia* drought resistance. 

### 2.4. Brassinosteroids Biosynthetic Pathway Is Involved in the Response to High-Temperature Stress

KEGG analysis, based on RNA-seq data, showed that brassinosteroids (BR) biosynthetic pathways are key pathways that are involved in the response to high-temperature stress. As presented in Figure 5, in the BR synthesis process, key genes that control the synthesis of intermediate products are significantly responsive to high temperatures. For this BR synthesis pathway, one DEG encoding *DEETIOLATED2* (*DET2*) and one DEG encoding *cytochrome P450 family 90D1* (*CYP90D1*) were significantly upregulated by high-temperature treatment; meanwhile, two DEGs encoding *DWARF 4* (*DWF4*), two DEGs encoding *CYP90A1*, two DEGs encoding *CYP85A1*, one DEG encoding *CYPA6*, and six DEGs encoding *CYP731A1* were significantly downregulated (Figure 5). All these results suggested that BR was important in response to high-temperature stress in *Pyrus betulaefolia*.

### 2.5. Transcriptome of Pyrus betulaefolia in Response to Both Drought and High-Temperature Stress

In order to understand the relationship between drought and high temperature for *Pyrus betulaefolia*, we performed an overlap analysis using the data of drought and high-temperature treatment. The Venn analysis showed that 880 DEGs respond to both drought and high-temperature stresses (Figure 6A). These expression patterns of those common DEGs at different treatment times were visualized with a heatmap, as presented in Figure 6B; here, a total of 294 DEGs were markedly downregulated, while another 155 DEGs were markedly upregulated by high-temperature and drought stress. A total of 205 DEGs were upregulated by high temperature and downregulated by drought, while 226 DEGs were downregulated by high temperature and upregulated by drought. However, the expression of some genes in “H_9 h_2” sample differed from those in the other two replicate samples “H_9 h_1” and “H_9 h_3”. In addition, the expression of some genes in “H_0 h_1” sample different from that in the other two replicate samples “H_0 h_2” and “H_0 h_3”. 

Subsequently, GO and KEGG analyses were performed to explore the function of DEGs. These DEGs could be divided into three main categories: biological process, cellular component, and molecular function, and most DEGs mediate catalytic activity, binding, the membrane components, and cell components, along with metabolic and cellular processes (Figure 6C). KEGG enrichment analyses found that the DEGs were mainly in the following six pathways: ribosome, pentose phosphate pathway, ribosome biogenesis in eukaryotes, photosynthesis, nitrogen metabolism, and glycolysis/gluconeogenesis (Figure 6D). Transcription factors are important regulators of the expression of functional genes in response to drought and high temperatures. In the overlap analysis, we identified many differentially expressed TFs that respond to both drought and high temperatures, belonging to the MYB, bZIP, WRKY, TCP, ERF, bHLH, Dof, ARF, and HSF families (Figure 7). In order to validate the expression of DEGs in the RNA-Seq data, the expression of these TFs which response to drought and high temperature were detected by RT-qPCR. The results showed that LOC103947119, LOC103950596, LOC103946681, LOC103966009, and LOC103960544 were significantly upregulated, while LOC103959917 was significantly downregulated by drought treatment. LOC103946681 and LOC103950544 were significantly upregulated, while LOC103927985, LOC103947119, LOC103959917, and LOC103966009 were significantly downregulated by high-temperature treatment. These results were consistent with the result of RNA-Seq (Figure 7B).

## 3. Discussion

Drought or high temperatures seriously affect the growth and development of crops. Plants have evolved various ways to cope with drought or high-temperature stress, including the regulation of morphology, response to cellular stress, hormone content, antioxidant content, and enzyme activity [1,15,19,23,24]. These stress responses are all based on regulation by corresponding genes. Such genes have been widely studied in model plants and mainly crops, including Arabidopsis, tomato, rice, maize, and wheat [25,26,27,28,29,30,31,32]. However, the corresponding pear genes in responding to drought or high temperature remain unclear. 

Previous study has analyzed DEGs that respond to dehydration in *Pyrus betulaefolia*. A total of 3483 DEGs were identified in the 6 h dehydration-treated samples compared with samples without treatment. These DEGs were enriched for photosynthesis, response to water, signal transduction, innate immune response, protein phosphorylation, response to biotic stimulus, and plant hormone signal transduction pathways. Furthermore, 637 TFs were shown to respond to dehydration [19]. The treatment method and materials used in this study were all different from those in our study, where we screened the DR seedlings of *Pyrus betulaefolia.* The DR seedlings were treated with drought treatment. The RNA-seq results revealed that 11,731 DEGs were identified as drought-response genes. These DEGs were enriched pathways that included plant hormone signal transduction, plant–pathogen interaction, glutathione metabolism, photosynthesis, nitrogen metabolism, and glycolysis/gluconeogenesis.

For high-temperature stress, HTR seedlings of *Pyrus betulaefolia* were obtained using the same method. We identified a total of 9639 DEGs that respond to high temperatures. These DEGs enriched the pathways that included protein processing, purine metabolism, and glycerolipid metabolism. It should be noted, however, that we cannot rule out the possibility of additional genes with altered expression at the time points. Moreover, we cannot rule out changes in protein levels that might not generally reflect the RNA concentration.

*Pyrus betulaefolia* is one of five primary wild species, which originated in China. *Pyrus betulaefolia* was used as rootstocks due to its high resistance to drought and high temperature, especially in the pear cultivation region of Northern China [33]. The resistance of *Pyrus betulaefolia* seedlings obtained by seed propagation is very different. Therefore, in this study, the number of *Pyrus betulaefolia* was screened for drought or high-temperature resistance. These resistant seedlings were treated with drought or high temperatures. Samples before and after treatment were collected to perform the RNA-seq. Through transcriptome sequencing, this study identified many DEGs that respond to drought and high temperatures. These works will provide valuable genetic resources and a theoretical basis for the molecular breeding of pears and provide the basis for the genetic improvement of pears.

It is well known that plant hormones are closely related to response to drought and high-temperature stress. Key genes of ABA signal transduction pathways, such as Pyrabactin resistance 1-like (PYL), Type 2C protein phosphatases (PP2C), and Sucrose non-fermenting 1 related protein kinases (SnRK), affect drought resistance [34,35]. In *Arabidopsis,* genes of the ABA signaling pathway significantly affected the high temperatures of plants [36]. Furthermore, genes associated with the ABA signaling pathway and the synthesis pathway significantly respond to high-temperature- and water-deficit-induced stress [37]. The auxin signal also regulates tolerance to drought or high-temperature stress [38,39]. *IAA5*/*6*/*19* positively regulate the tolerance to drought of *Arabidopsis thaliana* [40]. Overexpression of *OsIAA18* significantly improved tolerance to drought in rice [41]. Many studies have reported that IAA and ARF affect the response to high temperatures [39]. Jasmonates (JAs) have an important impact on plant response and stress resistance, and the function of JA signaling-related genes has been well reported. In rice, the overexpression of *OsJAZ1* significantly weakens the tolerance to drought compared with that of wild-type (WT) rice plants [42]. Transcriptome profiling analysis revealed many genes of ABA and JA signaling pathways expressed differently between plants and WT plants overexpressed with *OsJAZ1* under drought stress. In *Capsicum annuum*, *CaJAZ1-03* negatively regulates drought and ABA signaling [43]. However, the function and molecular mechanism of genes related to ABA, auxin, and JA signaling in drought or high-temperature tolerance was still unclear. In this study, several DEGs were identified that responded to drought or high-temperature tolerance. These results will provide valuable information for exploring the relationship between plant hormones and drought or high-temperature stress.

Brassinosteroids (BR), an important hormone, plays a crucial role in plant growth and development and resistance to adversity [9,44]. Several studies have reported the effect of BR on the response to drought stress. The exogenous application of Brassinosteroids significantly enhances the drought resistance of maize [45]. In Kentucky bluegrass, the drought treatment induced the activation of brassinosteroid-signaling kinase (BSK) [46]. Brassinazole-resistant (BZR) family TF were key genes in the BR signaling pathway. *TaBZR2* overexpression significantly enhanced the drought tolerance [47]. Furthermore, in spinach, several *BSK* genes were involved in responding to high-temperature stress [48]. The mutant plants with reduced BR synthesis exhibited significantly lower drought tolerance compared to plants with normal BR synthesis [49]. However, the mechanism behind the response of BR synthesis and signaling-related genes to drought and high-temperature stress in pears is still unknown. In this study, we identified many genes that respond significantly to drought or high-temperature stress. These results will provide a useful reference for gene function research.

## 4. Material and Methods

### 4.1. Plant Materials and Growth Conditions

The seeds of *Pyrus betulifolia* were collected from Shandong, Shanxi, Hebei, Anhui, Jiangsu, Hubei, and Jilin Provinces. The seeds of *Pyrus betulifolia* were then soaked in distilled water for 24 h before being placed in a box of expanded polystyrene foam filled with clean moist sand (5–10% moisture content); they were incubated at 4 °C for 40 days to achieve stratification. Then, all seeds were sown in the pots containing a mixture of soil: vermiculite = 1:1 (*v*/*v*). Seedlings were grown at 25 °C day/18 °C night temperature under 16 h light/8 h dark cycle, 70% relative humidity and 200 µmol m^−2^ s^−1^ light intensity.

After the roots were washed, the 20-day-old seedlings were transferred to Hoagland nutrient solution in which the *Pyrus betulifolia* seedlings were cultured and subsequently treated with 20% PEG6000. Most of the *Pyrus betulifolia* seedlings with PEG-simulated drought treatment wilted and turned black. Among these drought-treated seedlings, 30 seedlings with better growth, signaled by green and stretched leaves, were selected as drought-resistant type.

Next, 20-day-old seedlings were transferred to high-temperature incubators for high-temperature treatment. Similarly, most of the *Pyrus betulifolia* seedlings with high-temperature treatment have wilted and turned black. Among these high-temperature-treated seedlings, 30 seedlings with better growth statuses, signaled by green and stretched leaves, were selected as high-temperature-resistant type.

### 4.2. Treatment of Drought Stress and High Temperatures and Sample Collection

DR seedlings were transferred to Hoagland nutrient solution for 2 weeks. Twenty-four seedlings with consistent growth statues were used for drought treatment experiments. The samples were collected at 0 h, 12 h, 24 h, 36 h, 48 h, 60 h, 72 h, and 80 h after treatment with PEG6000 (20%), respectively.

The HTR seedlings were transferred to the substrate. HTR seedlings were treated with 47 °C after 14 days of growth on the substrate. We collected samples at 0 h, 3 h, 6 h, 9 h, 12 h, 18 h, and 24 h after HT treatment, respectively.

### 4.3. Total RNA Extraction, Construction of the cDNA Library, and Illumina-Sequencing

The total RNA of the leaves was extracted using TRIzol (Ambion, Austin, TX, USA), and DNase I was used to remove the genomic DNA residue. The quality of the RNA samples was evaluated using a NanoDrop Nd-2000 (Thermo, Fisher Scientific, Waltham, MA, USA). All RNA samples are used to construct libraries and sequences at Shanghai Majorbio Bio-pharm Co., using an Illumina NovaSeq 6000 System (Illumina, San Diego, CA, USA). The results of the sequencing were analyzed on a Major Bio Cloud Platform (https://cloud.majorbio.com/). Data were normalized by Fragments per kilobase per million (FPKM). FPKM was used to conduct gene expression analysis. Raw counts were analyzed for DEG using DESeq2 based on a negative binomial distribution with |log2FC| ≥ 1 and *p*-adjust < 0.05. The annotation and functional enrichment were used by The National Center for Biotechnology Information, Non redundant databases (Nr, http://ftp.ncbi.nih.gov/blast/db/FASTA/nr.gz), KEGG, https://www.kegg.jp/), and GO, http://geneontology.org/) databases. The GOATOOLS v1.2.4 (https://github.com/tanghaibao/GOatools) was used for the GO analysis with *p* value (FDR) < 0.05. The KEGG analysis were performed using the KOBAS v2.0 software (https://kobas.cbi.pku.edu.cn/home.do) with *p* value < 0.05. Heat maps were constructed using “TBtools”. All website accessed on 15 October 2023.

### 4.4. Quantitative Real-Time PCR Analysis

A measure of 1 µg total RNA was used to synthesize cDNA using the Prime Script™ RT Reagent Kit (Takara, Kyoto, Japan). The qRT-PCR was performed with the SYBR Premix Ex Taq polymerase (Takara) according to the manufacturer’s instructions. The relative expression of selected DEGs were detected by CFX Connect Real-Time System (Bio-Rad Laboratories, Hercules, CA, USA). Pear *PbrGAPDH* were used as the internal controls [50]. The 2^−ΔΔCT^ method was used to analyze the results [50]. Three biological replicates were performed for each sample. Primers used for RT-qPCR are listed in Appendix A.

### 4.5. Analysis of Antioxidant Enzyme Activities

The activities of SOD, POD, and CAT were measured in the leaves using a previously described method [51,52].

## 5. Conclusions

The present study conducted transcriptomic analyses using drought- or high-temperature-resistant seedlings. Several genes which have a significant response to drought or high-temperature stress were screened. These response genes were also enriched for the corresponding signal transduction pathways. The plant hormone signal transduction pathway and brassinosteroid biosynthetic pathways may be involved in response to drought stress and high-temperature stress, respectively. Further, the present study identifies many genes which respond to both drought and high-temperature stress. These results revealed potential genes which may affect drought or high-temperature resistance. Our study provides valuable information for the molecular breeding of pears for drought or high-temperature resistance.

## Figures and Tables

**Figure 1 plants-13-00309-f001:**
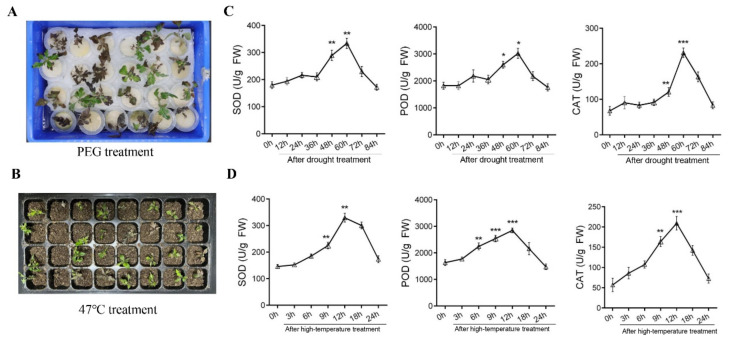
Isolation and analysis of partially drought-resistant (DR) and high-temperature-resistant (HTR) *Pyrus betulifolia* seedlings after treatment. (**A**) Representative phenotypes of DR and (**B**) HTR seedlings after drought (for 48 h) or high-temperature stress (for 9 h), respectively. (**C**) The detection of DR and (**D**) HTR seedling antioxidant enzyme activity under corresponding stresses, including superoxide dismutase (SOD), catalase (CAT), and peroxidase (POD) activities. Asterisks indicate statistically significant differences relative to the 0 h (*, *p* < 0.05, **, *p* < 0.01, ***, *p* < 0.001; Student’s *t*-test).

**Figure 2 plants-13-00309-f002:**
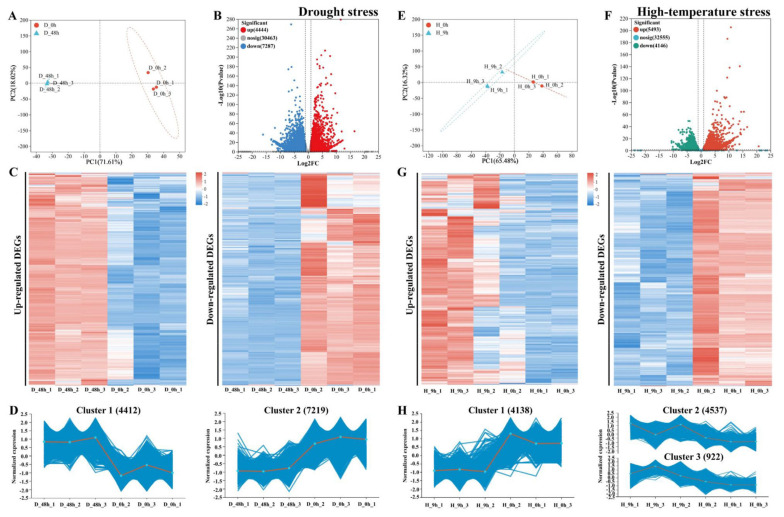
Identification of differently expressed genes (DEGs) from drought or high-temperature treatments. (**A**,**E**) Principal component analysis (PCA) of RNAs detected in *Pyrus betulifolia* seedlings before (D_0 h, H_0 h) and after being subjected to drought stress (for 48 h, D_48 h) or high-temperature stresses (for 9 h, H_9 h). PC1 means the first principal component and PC2 means the second principal component. (**B**,**F**) Volcano plots show the upregulated genes and downregulated genes of the comparison groups D_48 h_vs._D_0 h and H_9 h_vs._H_0 h, respectively. Red solid circles represent the samples (1, 2, 3) before drought (D_0 h) or high-temperature stress treatment (H_0 h), and cyan solid rectangles represents the samples (1, 2, 3) after drought stress for 48 h (D_48 h) and high-temperature stress for 9 h (H_9 h), respectively. Red dot indicates upregulated genes, blue or green represents downregulated genes, and gray means genes with no significant change. (**C**,**G**) Heat maps show the downregulated and upregulated genes in the comparison groups of D_48 h_vs._D_0 h and H_9 h_vs._H_0 h, respectively. (**D**,**H**) Hierarchical cluster analysis of DEGs in the comparison groups D_48 h_vs._D_0 h and H_9 h_vs._H_0 h, respectively. The blue lines in each group represent the gene expression pattern and the gray lines indicate the expression trend of all the genes in it.

**Figure 3 plants-13-00309-f003:**
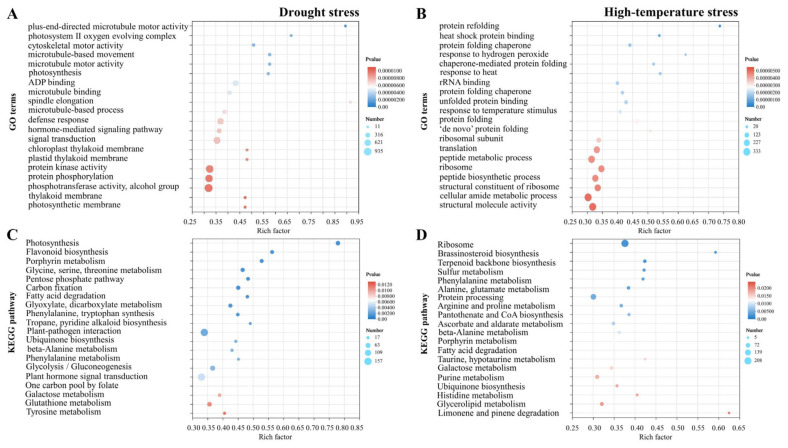
The functional enrichment analysis of the DEGs obtained from drought or high-temperature treatments. (**A**,**C**) The enriched Gene Ontology (GO) terms for DEGs identified from *Pyrus betulifolia* seedlings before (D_0 h, H_0 h) and after being subjected to drought stress (for 48 h, D_48 h) or high-temperature stresses (for 9 h, H_9 h), respectively. (**B**,**D**) The Kyoto Encyclopedia of Genes and Genomes (KEGG) enrichment analysis of DEGs isolated from *Pyrus betulifolia* seedlings before (D_0 h, H_0 h) and after being subjected to drought stress (for 48 h, D_48 h) or high-temperature stresses (for 9 h, H_9 h), respectively. The number of genes in each pathway is equal to the size of the dot. The dot color represents the *p*-value.

**Figure 4 plants-13-00309-f004:**
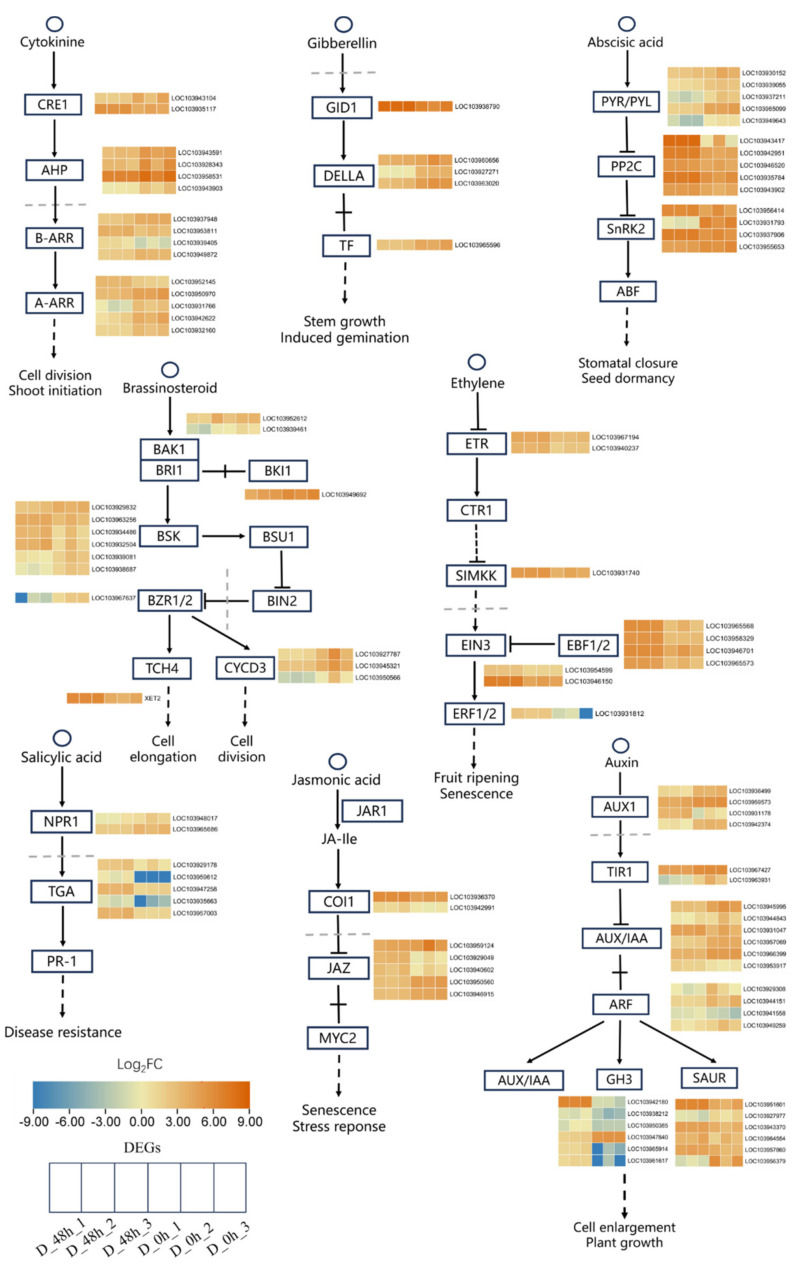
An analysis of DEGs involved in plant hormone signaling in *Pyrus betulifolia* under drought stress. The six boxes from leaf to right represent D_48 h_1, D_48 h_2, D_48 h_3, D_0 h_1, D_0 h_2, and D_0 h_3, respectively. DEG expression levels are normalized to log2 counts based on the fragments per kilobase of transcript per million mapped reads (FPKM) values and are presented in colors. The blue-to-orange gradient denotes a gradual increase in gene expression.

**Figure 5 plants-13-00309-f005:**
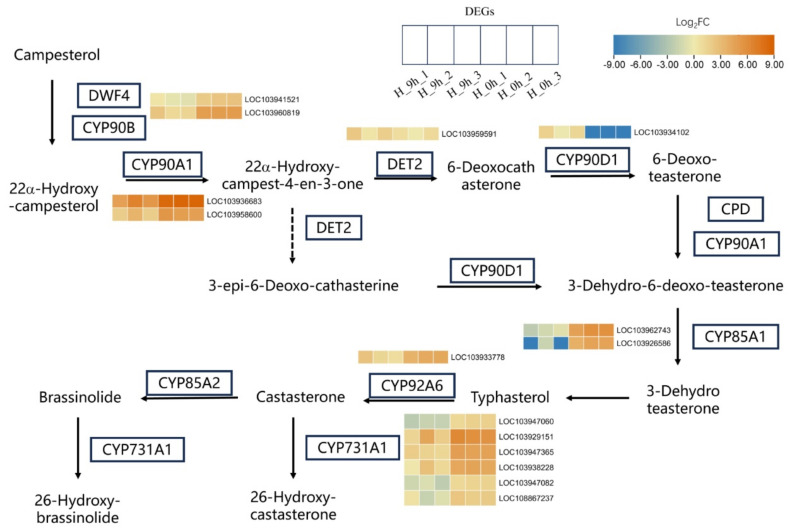
An analysis of DEGs involved in the brassinosteroids biosynthetic pathway in *Pyrus betulifolia* under high-temperature stress. The six boxes from leaf to right represent H_9 h_1, H_9 h_2, H_9 h_3, H_0 h_1, H_0 h_2, and H_0 h_3, respectively. The expression levels for DEGs are normalized to log10 counts based on the fragments per kilobase of transcript per million mapped reads (FPKM) values and are presented in color; the blue-to-orange gradient denotes a gradual increase in gene expression.

**Figure 6 plants-13-00309-f006:**
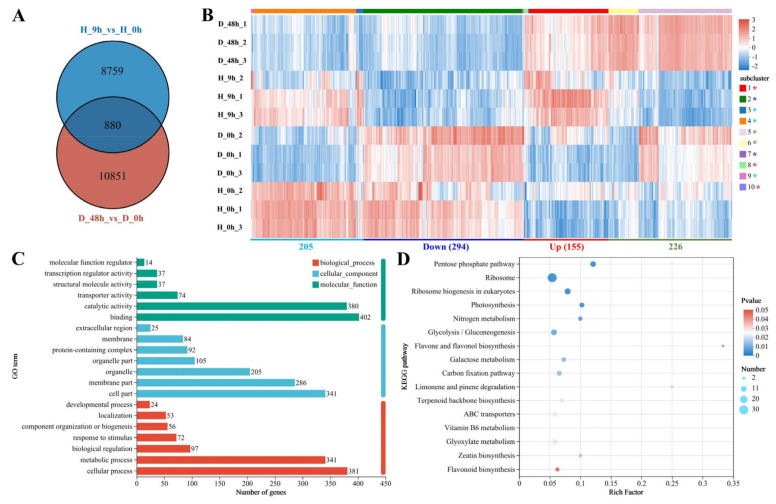
The overlap analysis of the differently expressed genes (DEGs) responding to drought and high-temperature stresses. (**A**) Venn diagram analysis shows that 880 DEGs respond to both drought and high-temperature stresses. (**B**) The heat map shows the DEGs that are downregulated and/or upregulated by drought or high temperature among these 880 DEGs. The clusters in the same group with a similar or an opposite expression trend in D_48h vs. D_0h, and H_9h vs. H_0h comparisons were decorated with the same colour asterisk. The enriched gene ontology (GO) terms (**C**) and The Kyoto Encyclopedia of Genes and Genomes (KEGG) enrichments (**D**) for these 880 DEGs. The number of genes in each pathway is equal to the size of the dot. The dot color represents the *p*-value.

**Figure 7 plants-13-00309-f007:**
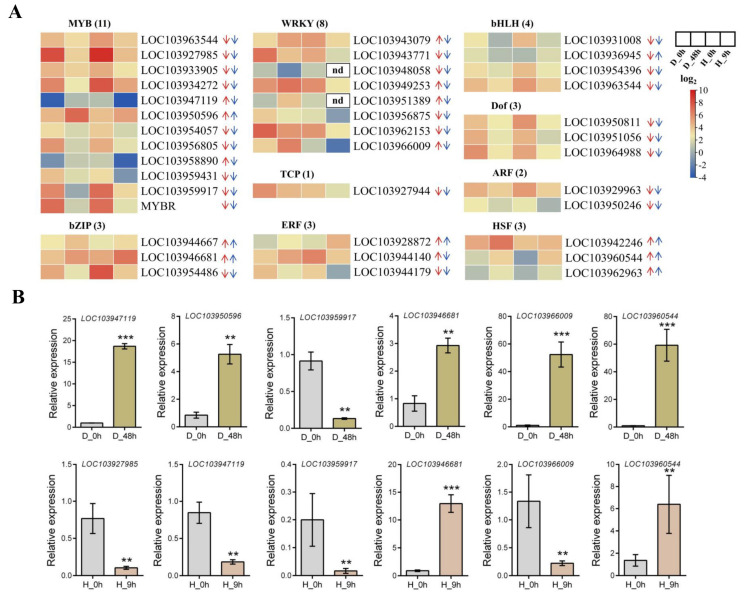
The main responsive transcription factors in the overlap analysis of the DEGs which respond to both drought and high-temperature stresses. (**A**) The heat map shows the TFs that are downregulated and/or upregulated by drought or high temperature. The red and blue arrows indicate the trend of changes in genes expression in D_48 h_vs._D_0 h and H_9 h_vs._HD_0 h, respectively. The up arrow indicates the upregulation and the down arrow indicates the downregulation by stress, respectively. (**B**) RT-qPCR was used to analyze the expression of some TFs before and after drought or high-temperature stress. Data represents mean ± SD of 3 biological replicates. Asterisks indicate statistically significant differences relative to the 0 h (**, *p* < 0.01, ***, *p* < 0.001; Student’s *t*-test).

## Data Availability

The RNA-sequencing data have been deposited in the National Center for Biotechnology Information (NCBI) with BioProject ID: PRJNA1044156.

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
