# Peer review of "Transcriptome Analysis Reveals Key Genes Involved in the Response of Pyrus betuleafolia to Drought and High-Temperature Stress"

_plants, 2024, doi:10.3390/plants13020309_

Round 1
Reviewer 1 Report
Comments and Suggestions for Authors
General Comments
It is not clear how the authors define “drought-resistant plants” and “high-temperature resistant seedlings”. What are the criteria for screening “drought-resistant plants” and “high-temperature resistant seedlings”? All of these should include detailed descriptions in the Methods and the Results.
The authors should include detailed descriptions on enrichment analysis for GO and KEGG in the Methods. What are the software/bioinformatics tools (versions) used for the enrichment analyses. What are the parameter settings for the enrichment analyses?
Specific Comments
Lines 32-33
I believe that “accumulation of reactive oxygen structure damage in the cell membrane and reductions…” should be changed in “accumulation of reactive oxygen species (ROS), structure damage in the cell membrane, and reductions…”.
Line 38
I believe that “plant stomata close leaves curl,” should be changed in ““plant stomata close, leaves curl,”.
Line 180
I believe that “Go analysis revealed…” should be changed in “GO analysis revealed…”.
Line 203
What does CRE1 stand for?
Lines 205
What does AHP stand for?
Line 205
What does CRE1 stand for?
Line 207
What does GID1 stand for?
Line 214
What does BAK1 stand for?
Line 217
What does BZR1/2 stand for?
Line 219
What do ETR and SIMKK stand for?
Line 220
What does ERF stand for?
Line 221
What do AUX1 and TIR1 stand for?
Line 222
What does ARF stand for?
Line 223
What does GH3 stand for?
Comments on the Quality of English Language
Okay.
Reviewer 2 Report
Comments and Suggestions for Authors
File attached

Comments on the Quality of English LanguageMinor English Language required
Reviewer 3 Report
Comments and Suggestions for Authors
Plants Manuscript #: 2772216
Authors: P. Ma et al., 2023
Title: Transcriptome analysis reveals key genes involved in the response of Pyrus betuleafolia to drought and high-temperature stress
The authors have presented molecular genetic approach to identify genes in the plant pear (Pyrus betuleafolia) to have changes in gene expression in response to 48 hours of drought stress and 9 hours of heat stress among a subset of both drought- and high-temperature-resistance varieties of pears, at the seedling stage. The goal is to identify key genes/proteins/pathways that help pears, an important fruit tree, to withstand these common and growing stresses. They used the common and appropriate methods, mostly RNA-seq-based procedures, for their study. There are some issues and questions I have about this, which I mention below.
There are several significant issues/questions I have about their manuscript/study that I feel need to be addressed prior to publication. These are as follows:
First, I was quite surprised that the authors both did not detect, or at least did not mention, the detection of gene expression changes in what are well-characterized heat stress response genes/proteins/pathways. These would include the very many Heat Shock Proteins (HSP’s) that also function as chaperones: HSP100’s, HSP90, HSP70, etc… These have been identified and investigated extensively (three quick references related to these):
Plant Biotechnol J. 2017 Apr; 15(4): 405–414. doi: 10.1111/pbi.12659
Cell Stress Chaperones. 2020 Nov; 25(6): 1071–1081. doi: 10.1007/s12192-020-01144-7
Plants (Basel). 2022 Dec; 11(24): 3410. doi: 10.3390/plants11243410
It has been known/reported for 20 plus years that the HSPs and their transcriptional regulation is controlled, in part, by the Heat Shock Transcription Factor (HSF). Interestingly, HSF is among the genes that the authors on this manuscript identified (and mentioned) as being altered in expression from their data, which is both expected and useful. But, that would suggest it even more that some if not many of the HSP genes would have altered expression (DEGs) in their data. However, even if none of the many HSP’s were not among the differentially expressed genes (DEGs) or the GO or KEGG identified pathways, the roles of HSPs is so commonly studied that at least mentioning that these were NOT among the differentially expressed genes would be noteworthy, and a bit surprising, and should be mentioned in the manuscript.
Second, none of the RNA-seq data have been independently confirmed. Virtually all papers that provide RNAseq added also test key genes using an independent RNA quantification method. Most often this is RT-qPCR (quantitative RT-PCR). It would not need to be for all of the DEG’s RNAs, but independent confirmation of a few is important. This is especially key if there is evidence of inconsistency and variance within the RNAseq data, which there is in this manuscript. For example, Figure 6B shows that among the three independent RNA replicate samples there is significant variance between replicates (compare sample “H_9H_1” to “H_9H_2” and “H_9H_3” AND compare “H_1H_1” to “H_0H_2” and “H_0H_3”).
Third, a general point that should be mentioned somewhere are two realities that are common issues with all RNAseq methods. The first is the fact that they selected time points (9 hours for heat and 48 hours for drought) to collect RNA for RNAseq and DEG analysis. They have to start somewhere, and they make a good argument for why they selected these time points. But, it is also quite possible that some key genes involved in either drought or heat shock have altered expression (DEGs) at different times. These could have dramatic changes in expression but at earlier times, say within the first hour or two of the stress, and if that is the case, their RNAseq would not detect these changes. This is always the issue no matter what the time point, unless one were to collect a very extensive (1 hr, 2, hr, etc….) time course, which is difficult for workload and cost reasons. My point is, the authors need to mention something to the effect that they “cannot rule out that some genes could have altered expression at time points that were not studied in this initial investigation and therefore would not be detected here”. The second is the fact that RNAseq only detects changes in the RNA amount, and that does not always correlate with equal changes in protein amount or protein activity changes. Again, a general statement to the effect that they “cannot rule out changes in protein levels that might not generally reflect the RNA concentration” would be useful.
Four, it would have been very helpful to compare the drought and heat resistant seedling data to drought and heat “sensitive” seedlings/varieties. This is because it is possible that whatever the genes or changes in gene expression in the “resistant” strains are, knowing which genes do NOT have those changes would be needed to pinpoint those genes with changes that distinguish “resistant” from “sensitive” varieties. An explanation as to why some “sensitive” strains were not tested for comparison is needed.
Five, further explanation in methods is needed, relates to details being needed for how the “selection” of the drought-resistant and high-temperature-resistant varieties were decided. The Results (Lines 103 and 119, respectively) mention that resistant seedlings were identified. The Methods (Lines 364-371) discusses the conditions and how long the seedlings were treated, but does not provide what “growth statues” (as stated in the Methods text). It just says samples were collected. What was the growth characteristics used for both drought and high temperature? Height? Biomass? Color of leaves? Further explanation of this is needed, so as to be able reproduce these same selections in the future.
I feel that these five main issues/concerns need to be adequately addressed before the work would be publishable.
There are also a number of smaller/writing/clarification issues, which I list by Line in the section below.
Abstract:
Line 16 : Write out full definition of DEG.
Line 24: mention some of the transcription factors with altered expression. It seems the HSF (Heat Shock Transcription Factor) should also be mentioned here, for it is a known, key TF in response to heat stress, as often reported (see key issue mentioned above).
Introduction:
Lines 33. Do the others mean “structural” damage?
Line 36: The “…heat stress, including in terms of ….” is awkward wording. Rewrite to be more clear.
Line 40: I would replace “body” with word “plant.” Term “body” not used to refer to a plant.
Line 59: The “…which hormone signal genes in ….” is awkward wording. Rewrite to be more clear.
Line 86: I feel the authors need to make it more clear to readers why their new study is distinct from the #21 reference, which also tested drought resistance in Pears. It should also be noted than in the Full Reference List at end (Line 453) the list of authors for Citation #21 is not correct. The name of first author is not correct.
Line 89: The “…seedings which resistant ….” is awkward wording. Needs a comma and “is” added, from how I interpret the meaning.
Line 97: This study “provides” (not will provide).
Results:
Line 103: As mentioned above, the authors need to provide more details about what plant and growth characteristics were used to select for the “resistant” seedlings. Height? Biomass? Number of leaves? Color? Etc…
Lines 110-112 AND 125-128. As mentioned above, with the 48 hr and 9 hr time selected, they authors need to state that there might be some genes with altered expression outside of these time points that these data would not detect. Note, this would especially be likely for key early regulators of these responses, for early regulatory factors often activate their response within short time period.
Lines 177-179: Why was the KEGG pathway of “photosynthesis” not mentioned in the text? It had the largest change and was the top listed category, but it was not mentioned at all in the text. Further, this was a category that was identified/published previously (#21) that had a significant change. So, to not mention it seems even more of error.
Figure 2: The text/font labels for the GO and KEGG categories are too small to read. The figure needs to reformatted to be readable. Further, for panels A, B, E, and F, the icon shape and color (which could not be read in key) should also be explained in the figure Legend.
Lines 196-227: Throughout this section the authors mention many gene names by their abbreviation and locus/gene ID number. Since these are the key genes with altered expression it seems worthwhile to define the genes and use the full name on first use, and not just abbreviation. Also, throughout this section there were a number of grammatical typos in which the verb-noun spellings did not match.
Figure 4: The six empty rectangles in the lower left corner of the figure with the sample names (“D_48h_1”, for example) needs to be explained better for reader to link that with the color pattern next to each gene. This was not immediately clear. Further, the last sentence of the Figure Legend is not clear to me, and the way I would read this, it seems backwards. That is, the “orange-to-blue” gradient would seem to denote a “decrease” in gene expression, since blue (according to the “Log2FC” key in the figure) has a “-“ (decrease) fold change while orange has “+” (increase) fold change. So going from orange (high RNA level) to blue (low RNA level) would be a decrease in expression. If that is not correct, please re-write to be clear.
Figure 5: Same issues as for Figure 4 above, with the six empty rectangles in the figure key and the “blue to orange” change showing what I interpret as a decrease in expression.
Figure 6B: As mentioned above, the sample “H_9h_1” seems significantly different than the other two replicate samples ““H_9h_2” and “H_9h_3”. If so, that should be mentioned somewhere. Likewise, “H_0h_1” seems significantly different than the other two replicate samples ““H_0h_2” and “H_0h_3”, and this should also be mentioned somewhere.
Lines 186-187: Here is where I first expected to see gene expression data on potassium-deficient stress impact on HAK gene expression, tested at the RNA level. But, not mention of this happens. Are there no gene expression (RNA seq) public data for Quinoa when grown under low potassium stress?
Line 270: Needs to read, “….the membrane components, and cell components…”
Lines 271-273: Why are the two KEGG pathways for Photosynthesis and Nitrogen Metabolism that both show high Rich Factor and reasonable number of genes involved NOT mentioned here. It is quite possible the Ribosome Biogenesis in eukaryotes pathway shows a change due to reduction in chloroplast ribosomes which also directly effects both photosynthesis and possible nitrogen metabolism, which in part happens in chloroplasts.
Figure Legend 7, Line 281: add “expression” after “genes” so that it reads, “…changes in gene expression in …” Otherwise it would imply mutations in the genes.
Discussion:
Line 292 The first sentence in this paragraph states that previous study has analyzed DEG’s in pears, yet this was not cited here. It should be cited, and I believe it would be #21, which as mentioned above has an error in the first author name in the Reference for #21.
Lines 301-302: Based on the provided data, this list should also include photosynthesis and nitrogen metabolism.
Line 308: sentence mentions high resistance, but not to what. Add “resistance drought and high temperature” to this, for clarity.
Lines 318-319: Define/write out full name of gene/proteins for PYL, PP2C, and SnRK, for the reader.
Line 322: Should be “auxin” not “autin”
Line 323: remove “s”, so it reads, “…IAA5/6/19 positively regulate the …”
Line 346: Define “BR” (most likely brassinosteroids?).
Materials and Methods:
Line 355: Define what type of media used in pots (soil, vermiculite, other?)
Line 355: Cite method or explain what was done for the stratification of seeds.
Line 358: Was the “nutrient solution” for growing plants in hydroponics of as a fertilizer of plants in soil?
Line 360-361: As mentioned previously, more explanation for what growth characters were used to select for the “resistant” seedlings is needed. As it reads, it is just “…better growth…” which is too vague. Please clarify and specific growth characters selected.
Conclusions:
In general, the Conclusions are very short and general so as to not be very informative. Also, in general the Conclusion would be written in present tense, so it would read “provides valuable …”
References:
Lines 453-454: Reference #21 has wrong first author listed, based on what I see in the actual paper. Correct author names for this citation.
As mentioned in my general statement, there is the well-studied and published topic of Heat Shock Proteins (HSPs) in plants that was not mentioned at all in the manuscript. It should be along with key citations added for this.
Comments on the Quality of English Language
See my general comments/suggestions
Round 2
Reviewer 3 Report
Comments and Suggestions for Authors
Plants Manuscript #: 2772216
Authors: P. Ma et al., 2023
Title: Transcriptome analysis reveals key genes involved in the response of Pyrus betuleafolia to drought and high-temperature stress
Second review of this manuscript, and the authors have done a generally good job in addressing many of my initial concerns, including the key scientific issues and missing RT-qPCR data, adding the many HSP genes that apparently are altered here, and adding language about not ruling out changes in times they did not test. I still feel that at some point they need to compare DEG between “resistant” and “sensitive”, for only with those data will they be able to identify the genes with DEG that directly cause the “resistance”. However, as they said in their comments and that I can accept, there are significant logistical and costs that are outside of their initial study.
There are still two issues that I feel need to be addressed before publication: 1) there remain some key, but easily fixable language problems, which I list below and 2) the Conclusion section still feels incomplete (note, another reviewer also felt it was incomplete). Once the writing grammar and language corrections are made, I could support it being publish. Note, I do not need to see the final edited version. The Plants Editor can confirm the suggested changes were made.
Introduction:
Lines 92. Still an issue with this sentence. Should read, “…seedings with resistant to drought and ….”. Not, “resistance of drought…”
Results:
Line 103: As mentioned above, the authors need to provide more details about what plant and growth characteristics were used to select for the “resistant” seedlings. Height? Biomass? Number of leaves? Color? Etc…
Lines 115-117. This section is getting closer, but still awkward wording. I would suggest, “However, there is the possibility that additional genes respond to drought but do so at earlier or later time points than what were tested in this study.” Then, I would move this new sentence to after the, “Therefore, we selected the 0 h and …the RNA-seq” sentence.
Lines 133-135: Similar to above. I would suggest, “However, there is the possibility that additional genes respond to high temperature but do so at earlier or later time points than what were tested in this study.” Then, I would move this new sentence to after the, “Therefore, we selected the 0 h and …the RNA-seq” sentence.
Lines 280-283: Awkward wording. Should be, “…sample differed from those in the other two replicate …. Same for lines 282-283.
Lines 298: Not sure why “Nitrogen Metabolism” is capitalized while all the other pathways were not.
Line 345: I agree with the general idea of this statement, but the wording is awkward. I suggest instead, “It should be noted, however, that we cannot rule out the possibility of additional genes with altered expression at time points…”
Materials and Methods:
Lines 408-409: awkward wording. I suggest, “…simulated drought treatment were wilted and turned black.”
Line 410: awkward wording. I suggest, “…with better growth, such as green and stretched leaves, were selected…”. Note, Materials should be written in “past tense.”
Lines 415-416: Same issue and suggested changes as for Line 410.
Comments on the Quality of English Language
Comments mentioned above.
